# Learning outcomes, learning support and cohort cohesion on a virtual field trip: an analysis of student and staff perceptions

Clare E. Bond[1], Jessica H. Pugsley[1], Lauren Kedar[1], Sarah R. Ledingham[1], Marianna Z. Skupinska[1],
Tomasz K. Gluzinski[1], Megan L. Boath[1].
[1] School of Geosciences, University of Aberdeen, Kings College, Aberdeen, AB24 3UE, UK

*Correspondence to*: Clare E. Bond (clare.bond@abdn.ac.uk)

**Abstract.** The rise of the virtual field trip was unprecedent in 2020 and 2021 due to the global COVID-19 pandemic. Virtual field trips aim to replicate the learning outcomes and experiences of actual field trips, by providing a digital alternative to in-field courses. They provide valuable opportunities for those unable to visit the field and alternative learning experiences for those that can. However, understanding their efficacy in terms of learning outcomes, the effectiveness of learning support offered, and cohort cohesion generally remains untested. Here we show how negative aspects of a virtual field trip both pre- and post-course are countered by positive outcomes in terms of breadth of learning outcomes and experience. As part of our analysis, we tested methods to mitigate barriers to inclusion and learning on a virtual field trip including internet connectivity and hardware access, use of printed workbooks, and limitations to interaction, support and cohort cohesion. Our results show that although negative perceptions, as evidenced through questionnaire responses, are dominant, with 71%- of the 27 pre-course respondents and 88% of the 21 post-course respondents commenting on these aspects across both student and staff cohorts; positive aspects of virtual field trips (43%-57%) also feature highly. Students show a positive shift in their perception of online teaching and learning over the course with positive comments moving from 19% pre-course to 71% post-course, whereas positive comments by staff are low pre- and post-course at 14%. Printed workbooks, staff: student ratios and interaction are received positively. Overall, we find that negative perceptions of virtual field trips pre- and post-course exist, but that both students and staff identify positive elements including breadth of learning outcomes, particularly regarding data synthesis and analysis. We suggest ways to learn from these findings to design virtual field trips that deliver effectively in blended learning environments for the benefit of all.

## 1 Introduction

Geological fieldwork in the years 2020 and 2021 has been significantly impacted by the COVID-19 global pandemic (Arthurs, 2021, Rotzein et al., 2021). In the United Kingdom this resulted in the country going into 'lockdown', an effective stay at home order, restricting travel and social interaction, from late March 2020 (Iacaobucci, 2020). Other countries were similarly affected. The result was no geological fieldwork could be undertaken, with many undergraduate programmes heavily reliant on summer fieldwork placed in jeopardy. For undergraduate students at the end of their penultimate BSc year this placed them, and staff, in the difficult position of missed credits from Easter and early summer field trips. With no likelihood of a summer mapping camp, or dissertation, students were on track to enter their final year with significant credit deficit and minimal field experience.

Virtual outcrops and virtual learning environments had slowly been developing over recent decades (Hurst, 1998, Tuthill and Klemm, 2002, Pringle et al., 2004, Trinks et al., 2005, Buckley et al. 2010 Çaliskan, 2011, Tibaldi et al. 2021), with increasing use and application in research (Casini et al. 2006;
Cawood et al., 2017), teaching (Tibaldi et al. 2020, Bond and Cawood, 2021) and conservation (Martínez-Graña et al., 2013, Pasquaré Mariotto et al., 2021). The effectiveness of virtual outcrops and virtual field trips had been, compared to actual in-field experiences, little evaluated, for an early example see Dohaney et al., 2012. But with almost 100% change in the sector from actual to virtual field trips overnight in early 2020 there are now a growing number of publications and papers in this
area (Mead et al. 2019, Whitmeyer and Dordevic, 2021, Bond and Cawood, 2021, Bos et al., 2021), of which papers in this volume are an example. At the University of Aberdeen, UK, in common with many global academic institutions, staff moved quickly to replace field training with virtual alternatives. In many respects Aberdeen was well placed with existing expertise and resources in virtual outcrop model creation and use, and with open access models that could be used by the broader geological community
(e.g., eRock; www.e-rock.co.uk, see Cawood and Bond, 2019, and v3Geo www.v3geo.com see Buckley et al., 2022). In addition to previous expertise in using online resources such as the UK virtual microscope www.virtualmicroscope.org (see Herodotou et al., 2018) and the virtual seismic atlas www.seisimcatlas.org. But concerns remained over the efficacy of multiple aspects of online learning and virtual field trips.

Rather than focus on the design and delivery of the geological elements of the virtual field trip, in this contribution we consider the issues of student and staff perceptions on learning outcomes, learning support and cohort cohesion, with a view to learning to inform future virtual field trip delivery. An initial key concern was internet connectivity and digital infrastructure, which was pertinent to the course
design and delivery, then following this the lived experience of students in using online resources and learning through participation in a virtual field trip.

We were first interested in ensuring effective course design and the impact on student learning and student satisfaction/dissatisfaction of the course arising from digital infrastructure. Internet connectivity
and digital infrastructure have been identified as a barrier to education with, implications for equality, diversity and inclusion (Laksana, 2021, Pearson and Reddy, 2021, Devkota, 2021, Ochieng and Gyasi, 2021), and although many examples focus on developing countries, similar issues are found in the most advanced global economic countries (e.g., Sanders and Scanlon, 2021). Internet connectivity is essential for the delivery of online courses and for the real time use of 3D virtual outcrop models during virtual
field trips. In this regard digital resources are particularly pertinent to the delivery of virtual field classes as they can require significant internet speeds for live streaming. The rendering of 3D geological models also often requires 'higher-end' graphics cards, causing specific challenges. Such digital capacities in education are an issue identified and outlined in detail by Abduvakhidov et al. (2021). Our second focus, student perceptions of the challenges they would face in completing a fully online virtual
field course; was chosen as perceptions have important implications for learning ability and learning outcomes. Evidence from pedagogic studies show that student concerns around learning environments, cultures and other stresses have an impact on learning ability and gains (Hess, 2002, Christie et al., 2008, Xia, 2009). Effectively being placed in an unfamiliar environment increases the mental load and

reduces capacity for engagement in learning. Pre-fieldwork anxiety has been demonstrated for students who are, for example, unfamiliar with fieldwork and/or the outdoors, or are concerned about the use of shared facilities on a field trip. (Boyle et al., 2007, Stokes and Boyle, 2009). Although familiar in one sense, working from home during COVID-19 was unfamiliar and uncertain, as was the delivery of a new and alternative course in a novel format using new software at short notice. Novel space has the potential to distract student participants from achieving the learning outcomes of the virtual field trip (Orion and Hofstein, 1994). The online environment also brings different challenges to learning including lack of connections and development of a learning community, sensu Tinto (2008) and a positive emotional setting for learning (Cleveland-Innes and Campbell, 2012). We refer to these aspects as cohort cohesion (as described by Wathington et al. 2010). We were interested in the students' perceptions of how their learning in this unfamiliar environment was affected. Similarly, our third focus considered, staff perceptions of the programme and delivery challenges. We were interested to see if student perceptions pre-course mirrored those of staff and if likely challenges identified by staff and associated mitigation strategies were apparent and effective. As well as evaluating how perceptions of staff and students changed over the programme.

To evaluate the three foci outlined we designed three online surveys (see Supplementary Material). Here we reflect on the outcomes of the three surveys, discussing the implications for learning and teaching in new formats, drawing out the potential benefits and challenges of different approaches and how staff and student perceptions changed.

**2 Designing the Course**

After 'lockdown' was announced a group of staff worked as a team, over the subsequent six-week period, to develop a five-week training program that would directly replace a two-week field trip. The original field trip was designed for penultimate year undergraduate students, in the NW Highlands of Scotland, and had previously acted as precursor to their final year six-week individual mapping dissertation. Significant consideration was given to the length and format of the replacement online training, with a longer programme eventually favored for inclusion purposes (e.g., students ill with COVID, students working in critical supply chain jobs, and students with caring responsibilities (schools and other facilities were closed)). This allowed a format of two recorded 'kick-off' sessions a week, essentially 10 work elements over the five-week period, with drop-in sessions at various times, including some evening sessions, to accommodate the diversity of individual situations, and availability, within the student cohort.

An initial exercise was undertaken amongst the staff team to consider the learning outcomes of the 'actual' field trip and how these mapped onto the possibilities for delivery in a virtual environment. The learning outcomes for the original field trip were: seeing rocks in their context in the field, making detailed observations of rock outcrops and fabrics, collecting structural data, completing sedimentary logging, interpreting and analysing structural data, interpreting and analysing sedimentary logs, building and maintaining a field notebook, field sketching, interpretating field observations to make predictions, interpretating field observations to build a geological history, synthesising datasets to create maps, cross sections, evolutionary understanding and analysis of palaeoenvironments. Through the series of

activities designed the staff team felt that all the learning outcomes could be achieved, apart from physically *'to observe rocks in their context in the field'*; although this could be done virtually. Alongside the geological learning outcomes was consideration of how academic and peer support could be used in a virtual environment to best achieve the learning outcomes and build cohort cohesion to aid learning. These latter elements were in many ways more challenging; the logistics of effective learning strategies with the external factors and uncertainty related to delivery during the early weeks of the COVID-19 pandemic, including a five-week long programme with potential for fragmented learning and limited cohort interaction were concerns. Mitigations to address these concerns were built into the programme design. Further concerns included the internet connectivity of students and staff to attend and deliver the programme.

## 2.1 Internet Access

The issue of the potential impact of internet connectivity was addressed immediately to inform course design and delivery. An initial questionnaire was conducted to test students' internet connection speeds: download-speed, upload-speed and latency (through a linked testing service), to determine what computer hardware they had access to, and to find out about their daily availability over the duration of the course. We were also conscious that the students would be facing potentially new and additional challenges during the course and included a free form question entitled 'Other Issues' in which we asked students to *"highlight any other issues that you feel you may have in completing the work (e.g., childcare commitments, key-working, lack of quiet space/time to work - note that these are just examples, this list is not exhaustive). Please give as much information as possible as to how any issues raised will likely affect you."* This initial questionnaire was completed by all students on the course. Students participating in the course were based in locations across the UK and Europe. Tests of the students' internet quality (download-speed, upload-speed and latency) established that the overall quality of internet in the student cohort was poor compared to published data from Speedtest (www.speedtest.net, accessed 05/2021, with results obtained for global averages in May 2020) and similarly from Ofcom for the UK (www.ofcom.org.net; accessed 05/2021, with results obtained for 05/2020) Figure 1a). The average UK upload-speed in May 2020 falls out with the upper quartile of student speeds and for download-speeds the UK average at that time was bettered by the only student with an ether-net cable who had the fastest download speed in the cohort. It is worth noting that students were asked to test internet connectivity at a time during the day when they were likely to be attending the programme. We also recognized that home-internet at the time was under significant stress, with the potential for several working adults on video-calls during the daytime on one home-internet connection.

We also monitored connectivity during the live sessions, so that we could react dynamically to issues. It was clear from this monitoring that some participants had issues with connectivity, identified by a red poor connection signal within the virtual classroom environment used for delivery. Evidence from the virtual classroom software suggests that these problems were however very limited with most students connecting just once to live sessions (Figure 1b). The data presented in Figure 1b) is for 22 live sessions over the duration of the five-week virtual field trip. We assume that those joining once or twice are joining and leaving sessions at will, whereas those joining 3 or more times are having connectivity or software issues. Note that of the 180 total joins to sessions 155 were single joins and 18 were double

joins, with only 7 joins falling into the 3+ category. This also implies that for students who did on occasion experience issues, that these were not persistent over the virtual field trip.


A series of mitigation measures were designed into the course to minimize the potential impact of internet connectivity issues, as well as the effectiveness of the hardware which the students had access to. Each live introductory session was recorded for student access after delivery. Similarly, question and answer sessions were also recorded. The only sessions not consistently recorded were smaller group

sessions led by PhD students. A printed work booklet was sent to all students in advance of the course, this contained a timetable of activities, each sessions explanatory sheets, information on formative and summative assessments, and session material e.g., detailed photographs and maps. This helped mitigate potential internet and hardware issues, for example, to render 3D models and visualizing images; and given that most of the students were working on laptops, with a single relatively small screen, it allowed

them to refer to material whilst also seeing the staff in a live video stream or looking at a virtual outcrop model. Some sessions required: Google Earth, the session instructions (PDF format), virtual outcrop models and detailed photographs, so having access to some of that material in hard copy was useful, ensuring that more than one element could be considered at the same time.

**2.2 Student availability and other issues**

There was a range in student availability with some students under-taking key-worker roles in critical service sectors whilst others were generally available. Other issues identified by the students were: Wi-Fi variability, childcare and a quiet working space, these issues were compounded by many students having returned to their family homes with multiple adults working at home online, and/or the presence

of younger siblings. Most of the mitigation measures that were put in place were around the length, frequency and timetabling of sessions for the programme and the availability of session recordings for all, with clear supporting written explanations. The smaller group sessions were timetabled so that all students at least had live access to these sessions. The purpose of the small group sessions was in-part to allow a 'safer' space (e.g., Gayle et al. 2013) in which students could ask questions in smaller groups

and to a PhD student, to breakdown potential issues around fear of speaking-up in the larger full-class setting and to more senior staff members. Although, a 'there's no such thing as a silly question' philosophy was imbued such mitigations can be hard to permeate and to be accepted within student cohorts.

**3 Eliciting Perceptions**

**3.1 Survey Design**

The main dataset analysed in this paper was sourced from two sets of online questionnaires completed by the students and staff participating in and delivering the course. Participation in the questionnaires

was voluntary and the University of Aberdeen ethical procedure was followed. The aim of the questionnaires was to inform our over-arching research questions:

1.  Did the participants and staff perceive that the learning outcomes were achieved?

2. Were effective measures put in place to support the learning outcomes and delivery?

3. Did the participants and staff perceive the peer-peer and academic support to be effective?

The first 'pre-course' questionnaire focused on participants' perceptions of the: i) learning outcomes of the trip, (ii) learning support (i.e., peer-peer and academic-student interactions), as well as some of the iii) logistics and challenges associated with online-distance learning, particularly virtual field trips. The questionnaire design and the statements for which perceptions were elicited were informed by the original field trip learning outcomes; and further developed by staff discussions during the design of the programme that reflected concerns with delivery of the learning outcomes and associated factors such as cohort cohesion and the logistics and issues of virtual field trip delivery. The full set of questions in the survey can be found in the supplementary material. All the questionnaires eliciting perceptions were answered anonymously.

The questionnaires started with two questions that elicited student and staff perceptions using an open text box response on the positive and negative aspects of a virtual field course. The questions were: "*What do you think the positive aspects of a virtual field course might be?*" and *"What do you think the negative aspects of a virtual field course might be?".* Participants were then asked to answer to what extent they agreed with 25 statements (Supplementary Material) using a likert scale response between 1 and 5, corresponding to how much they agreed with the statement. We chose a mixed methods approach with a likert scale to allow for easy analysis of the survey, but with an open text box in which participants could add qualitative statements to elaborate on their quantitative answers. The open text allowed participants to highlight any areas they felt were important but that had not been raised by the questionnaire statements. The student and staff questionnaires mirrored each other and were simply phrased for participation and delivery respectively. The second 'post-course' questionnaire was circulated after the course and focused on the same statements as presented in the 'pre-course' questionnaire, but from the new perspective of having completed the course or, for staff, having delivered it. It is important to note that not all student participants completed the pre-course questionnaire in advance of the course, and that some students returned theirs after the first week of the course.

**3.2 Survey Analysis**
There were 22 students enrolled on the course and 11 staff members consisting of 6 full-time staff members and 5 PhD student demonstrators. Different staff were involved in delivery for different weeks. For the pre-course questionnaire there were 20 student respondents and 7 staff respondents; for the post-course questionnaire there were 14 student respondents and 7 staff respondents. There were a couple of questions not answered by all respondents these were: Pre-course: 'Building and maintaining a field notebook', not answered by one staff member; Post-course: 'Collecting structural data', not answered by one student; and 'I think the ability for students (me) to engage in and complete work on the Virtual Field Trip will be affected by caring responsibilities', not answered by one student. Likert scale answers to the statements by all responding participants were summed, with median and inter-quartile ranges calculated and plotted in box and whisker format to show the range in perceptions of

staff and students pre- and post-course to the 25 elicited statements. We chose to present our likert data in box and whisker format, as this shows the median, minimum and maximum choices as well as the interquartile range, following Holzer et al. (2013) and Gregory et al. (2022). The full set of ordinal responses are presented in the supplementary material. In contrast, the free text comments required coding before analysis. The initial codes were developed by two of the author team, who defined a set of correlative codes that spanned the range of respondent opinions, both positive and negative. The coding was designed so that multiple codes could be assigned to each comment to capture the breadth of respondent perceptions during coding. Codes were then checked by the full author team to ensure they were relevant, and their descriptors were explanatory. Detailed reflection and refinement resulted in a final set of ten codes and associated descriptors (Table 1), that captured the critical elements of participants' comments. Codes were not explicitly designed to be paired. However, eight of the ten codes were effectively positive and negative pairs. These were: accessibility, equality, inclusion; time management; aspects of online teaching and learning; aspects of the virtual field trip. The codes were then applied to the qualitative free text responses by the author team, as three pairs and one individual to generate a set of codes for each participants' free text. The four sets of codes were then compared, discrepancies were minor. The full-author team met and reviewed the codes and agreed on a final set of codes for each free text response.

## 4 Perceptions

First, we consider the responses to the quantitative statements and then go onto consider the qualitative free text of participants in response to the first two questions posed and the open question option. The full dataset is provided in the linked data repository and all questionnaires in the supplementary material.

### 4.1 Quantitative Statements

Numerical responses were collated for each statement and plotted as box-and-whisker diagrams. Responses are collated into the three key themes: learning outcomes, peer-peer and academic support and logistics. The key findings are shown in Figures 2, 3 and 4 respectively, and grouped by statement to enable comparison between student and staff response, and pre- and post-course differences. Since the questionnaires were anonymised, changes in individual opinion could not be tracked, but by analysing the median, and interquartile ranges (IQR) of the sets of responses it is possible to interpret the collective perceptions of the two cohorts (students and staff) both pre- and post-course, and therefore track changes in perceptions with time. Differences between student and staff perceptions were observed for certain statements, the key observations are described below. It should be noted that one student responded with the most negative option for all the learning outcome statements; from their response to the open question, it was clear that this student was frustrated at not being able to go in the field, and we interpret their responses as reflecting this frustration. Often this negative response is shown as an outlier (a dot) in the box and whisker plots. Despite the low number of participants, particularly in the staff cohort, we ran a Mann-Whitney U test (Mann and Whitney, 1947) to ascertain significant differences in perceptions for the: students pre- and post-course, staff pre- and post-course and between students and staff pre-course and students and staff post-course, the full results of the Mann-Whitney U test are available in our data repository. The perceptions that show statistically

significant different at p<0.05 and in one case p<0.01 are highlighted in figures 2, 3 and 4, and
discussed in the results.

### 4.1.1 Learning Outcomes

Key responses to the learning outcomes statements are shown in Table 1. Responses of 1 reflect an
opinion of unlikely to be achieved, whereas 5 is likely to be achieved. The statement *"students are
likely to see rocks in their context in the field"* (Figure 2a) was typically met with neutral responses
prior to the course, with students and staff scoring the statement with a median response of 3.7 and
having the same IQR of 2-3. Post-course, the IQR increased for both cohorts, 2-4 for students and 1-4
for staff. The median student response decreased slightly to 3.5 whereas staff perception decreased to 2.
For the statement *"students are likely to make detailed observations of rock outcrops and fabrics"*
(Figure 2b), pre-course the students and staff had median scores of 3 and IQRs of 3-3.75 (students) and
2-3 (staff). Post-course the IQR for students narrowed to 2.5-3, whilst the staff IQR increased 2-4. The
median responses remained consistent at 3.

There was diverse opinion both pre- and post-course for students and staff with regard to the *"ability of
students to complete sedimentary logs"* (Figure 2c), with the IQR spanning 2-4 in all response sets, but
most participants agreed that the "*interpretation and analysis of sedimentary logs*" would be possible
(Figure 2d), with an IQR of 4-5 pre- and post-course; staff opinion remained unchanged with a median
of 4 pre- and post-course, whereas the student median increased from 4 pre-course to 5 post-course .
Similarly, students and staff were optimistic about the "*interpretation of structural data*" (Figure 2e)
pre- and post-course, the IQR was 4-5 with median scores of 4.5 (students) both pre- and post-course
and 4 (staff) pre-course and 5 (staff) post-course; similar too were responses to the "*interpretation of
field observations to make predictions*" (Figure 2f) with an IQR of 4-5 both pre- and post-course for
staff and students. Note that these statements do not refer to the method of data collection, so
the responses do not refer to the acquisition of field data, only the analysis and interpretation of that
data.

Students and staff both predicted that they would be able to "*build and maintain a field notebook*"
(Figure 2g), IQR 4-5 (students) and 3-4.25 (staff) in the pre-course questionnaire, there is however a
significant difference for p<0.05 (Mann-Whitney U test giving a U of 24.5 (U at p < 0.05 is 27)),
between student and staff perceptions pre-course with students significantly more confident than staff
that they will be able to build and maintain a field notebook. The post-course perceptions between the
two cohorts are not significantly different, the IQR of 3-5 broadening for students and decreasing
slightly, 3-4, for staff. The student median was 5 both pre- and post-course, whilst the staff median
increased from 3 pre-course to 4 post-course both students and staff felt it likely that this learning
outcome had been achieved post-course.

In similarity to the question on building and maintaining a field notebook, *'field sketching',* was thought
likely to be achieved by the student cohort pre-course, with a median of 4 and IQR of 4-5. For staff pre-
course the median was 3 and the IQR 2-4.25. These are statistically different, the U for p<0.05 is 34 and

the U returned was 25.5. Post-course the medians remain the same, but the IQR for the students increases to 3.75-5 and decreases for the staff to 3-4. These are not statistically different.

Students were unsure about their potential "*ability to interpret field observations to build a geological history*" (Figure 2i) prior to the course, with responses across a range of 2 to 5, an IQR of 3.25-4.75 and
a median of 4. Post-course, the IQR range narrowed 4-5, and shifted with a median score of 5. The Mann-Whitney U test showed this to be a significant change (giving a U of 25.5 (U at $p < 0.05$ is 34), with students perceiving that this learning outcome was more likely to have been achieved post-course. Staff responses showed a similar, but not statistically significant, trend with a narrower IQR post-course shifting from 3-5 pre-course to 4-5 post course and a post-course median of 5. Resulting in very similar
distributions for staff and student cohort's post-course. For a similar statement on being able to "*synthesise datasets to create maps, cross sections, and interpretations*" (Figure 2j) students and staff had a neutral to positive opinion pre-course with an IQR of 3-5 (students) and 3-4 (staff). Post-course the IQR was 4-5 for both staff and students and the median score increased from 4 pre-course to 5 post-course (students) and remained at 4 for staff.

### 4.1.2 Peer & Academic Support
Questions on peer and academic support were scored from 1 no to little impact, to 5 significant impact. There was diversity of opinion within the student and staff cohorts pre- and post-course on whether peer-to-peer learning and cohort cohesion would be/had been impacted compared to an actual field trip
(Figure 3a and b). For the question "*peer-to-peer learning will decrease compared to an actual fieldtrip*" the IQR changed from 4-5 (students) pre-course to 2.75-5 (students) post-course, with a consistent median of 4 pre- and post-course. Staff had a greater IQR of 3-5 pre-course and a median of 4. Post-course the staff IQR range remained the same, but the median score decreased to 3.

Prior to the course, students were unsure as to the "*level of academic support that they would receive as individuals*" (Figure 3d), with a median value of 3, an IQR of 1.25-3 and a full response range of 1-5. The range of opinions on completion of the course was still 1-5, but the IQR had shifted to 2.75-5, with the median score remaining at 3. For staff the IQR was consistent pre- and post-course 2-4 with the median shifting from 3 pre-course to 4 post-course. A positive shift in perception was seen from pre- to
post-course in terms of "*academic staff support for the group as a whole*" (Figure 3e) with median scores for students and staff rising from 3 pre-course to 4 post-course. The positive change in student perception was significant in the Mann-Whitney U test, with a U of 71 (U at $p < .05$ is 83).

### 4.1.3 Logistics
Regarding the length of the course, students and staff responded to two statements comparing a five-week period of distributed learning, with a 10-day intensive course, one question focusing on the benefit to them as individuals and the other on the benefit to the group. A score of 1 correlated with a bad perception and a score of 5 a good perception. There was a broad spread in response to both questions (Figure 4a and b). But both cohorts, students and staff, felt that the extended five-week time period was
"*beneficial to individual students*", with pre-course median scores of 3 (students and staff) and 3.5

(students) and 4 (staff) post-course. For the *"benefit of the group as a whole"*, median scores were consistent pre- and post-course with scores of 4 (students) and 3 (staff).

Students and staff had quite different perceptions of how much caring responsibilities would impact the students' ability to engage in and complete work (Figure 4c). Both the pre-course student and staff perceptions and the post-course student and staff perceptions are significantly different. Pre-course the Mann-Whitney U test gave a U of 12 (U at $p < 0.05$ is 34 and at $p<0.01$ U is 24), for the difference between student and staff perceptions. The students had a median of 1 pre-course and the staff a median of 4. IQRs were 1-1.5 (students) and 3-5 (staff), post-course the student median stayed at 1 with an IQR of 1-2.5. For the staff the post-course median and IQR was 3. Post-course the Mann-Whitney U test, for the difference between student and staff perceptions, gave a U of 13 (U at $p < 0.05$ is 20 and at $p<0.01$ U is 13).

Students predicted a range of outcomes regarding *"internet issues affecting progress"* (Figure 4d), with scores of 1 indicating not much impact and 5 a lot of impact. The full range of responses, 1-5, were submitted by students' pre-course, but the median value shifted towards little impact, from scores of 2 (staff and students) pre-course to 3 (staff and students) post-course.  Opinion on the value of a printed workbook were generally positive (Figure 4e-g). Scores of 1 indicated that it had not really helped, and 5 yes, a lot of help. Students' opinion was broad as to the usefulness of a workbook in terms of *"finding a quiet space to work"* in advance of the course (Figure 4e) with an IQR 2.25-4, and a median score of 4, whereas staff more consistently believed it would be useful (IQR 4-5), also with a median score of 4. Post-course median scores all remained at 4 (students and staff), but with an IQR spanning 1-5 (students) and 3-4 (staff). In terms of a workbook allowing *"students to reflect on their work away from a screen"* (Figure 4f), both students and staff agreed that this was true with positively skewed and narrow IQR ranges across pre- and post-course questionnaires, and median scores pre-course of 5 (students and staff) and post-course 5 (students) and 4 (staff). Students believed that the workbook would *"provide a resource for future reference"* (Figure 4g) (median scores of 5 pre- and post-course); staff scored this statement with a median score of 4 pre- and post-course.

## 4.2 Qualitative Statements

The questionnaire asked for long text answers to two questions, one on the positive aspects of a virtual field course the second on negative aspects. At the end of each of the questionnaires there was also free text space for respondents to add anything additional that they felt had not been covered in the preceding statements. Seven staff members involved in the course design and delivery provided free text comments before and after the field trip, along with 20 of the participating students, pre-course, and 14 post-course. The free-text responses highlighted issues ranging from technical aspects of virtual environments and field trips through to learning outcomes and experiences. Many provided context and reasoning for their scoring responses to the preceding questions, with free text responses clearly being led, to some extent, by the preceding quantitative questions. Ten thematic codes were created after analysis of the free text (see Table 1 and 3.2 Survey Analysis). These thematic codes naturally fell into positive and negative categories, and we used this classification (see Table 2, in which the full set of coded data is summarized). Figure 5 shows radar plots of the same data to visually represent the pre-

and post-course perceptions of students and staff respectively (Figure 5a and b), and comparison of staff and student perceptions pre course (Figure 5c) and post-course (Figure 5d). The radar plots are 'split' vertically with positive aspects on the right-side and negative aspects on the left-side. We describe the results of each radar plot in turn.

### 4.2.1 Pre- and post-course student perceptions

The radar plot of pre- and post-course student perceptions (Figure 5a) shows little change in student perceptions on the aspects that they raised over the course. Most of the change lies within the positive half of the plot, notably around positive aspects of online teaching and learning with 71% of respondents mentioning this after the course in comparison to 19% pre-course. Phrases used by student's post-course include aspects around the ability to ask questions and not be affected by the weather: *"Directly asking Q's on sessions easily", "No rain!!"* and the ability to engage effectively with staff: *"Individual engagement with professors during calls and being able to ask a lot of questions".* Negative aspect of online teaching and learning complement this with a decrease from 63% pre-course to 50% post-course mentions, but there were still clear perceptions of having missed an opportunity, in the words of one student *"Relationships between classmates and our teachers are made on field trips, and there is nothing like it to drop barriers and get people out of their comfort zone. These are the stories we take home with our degrees, and its these memories we will cherish over any qualification we achieve",* and recognition that some aspects of an actual field trip cannot be replicated: *"Lack of the field experience, physical interaction with rocks and ability to interact and ask questions/discuss things with examples to which you can point at", "Not being able to look at the outcrop in person, look at any features you perhaps cannot see in enough detail in images", "Nothing can replace actually seeing these outcrops in person. Being able to touch the rocks and see the entire setting of a location, being able to appreciate its beauty in real light and feel the enormity of what was going on with the geology from a 1st hand perspective. doing geology this way is why I enjoy the subject so much, I don't get the same gratification from just analyzing data provided. Also there is no chance you might happen upon something new doing field work in this way. Being able to contribute to the subject by finding something new for the first time must be really exciting....".*

Time management also shows a positive shift post-course from 25% pre-course to 36% post-course. Students enjoyed the flexibility in study time and ability to study at their own pace: *"Flexibility, can do the coursework when it suits you", "Tidy field notebook and the ability to move at your own speed as you may be rushed in the field", "Learning at your own pace", "You have a while to actually figure and process information",* but this is balanced by negative mentions which increase from 19% pre-course to 36% post-course for example: *"On the flip side, the fact that the virtual course was spread out over a much longer period of time was slightly annoying as well", "sometimes too much time to think about ideas, therefore causing confusion or over complicating things".* Mentions of negative issues related to software/hardware and Wi-Fi problems also increase from 25% pre-course to 36% post-course, for example *"Wi-Fi problems & the quality of image resolution that will load on the programs", "the course always required internet access to complete the task",* and *"Some of the programmes could not run on my computer, I did not have access to computers with higher processing power."*

The students also recognized that their views were influenced by the ongoing pandemic and the additional challenges this raised: "*I think that a lot of this, along with lockdown, is a psychological challenge in the respect that the actual act of going into the field; the build-up and pre-departure information, packing etc.. all help to push the brain into a place where it focuses on the task at hand and you can immerse yourself within the geology fully. Being at home doesn't force yourself into that*
*place and it's difficult to try to get your brain into "geology-mode" as it were. I'm sure all the staff are aware of this but I think personally, it's something that has become apparent completing the first few virtual assignments.*" and "*We all had to do this during a global pandemic (staff included), without our friends, isolated from our usual coping mechanisms e.g. gym, uni routine, being in Aberdeen, enjoying the great outdoors. So this is a particularly challenging time for you as staff and us as students to*
*complete & set a virtual field trip.*"

**4.2.2 Pre- and post-course staff perceptions**
Figure 5b illustrates the changes in staff perceptions pre-and post-course. Staff perceptions show some changes pre- and post-course. Comments on the negative aspects of the virtual field trip are dominant
and increase slightly over the course from 71% to 86%, with phrases such as "*Specific field techniques are not developed*", "*Difficulty applying digital-learnt skills to real-world scenarios when normal fieldwork practices recommence*" and "*Some field skills cannot be replicated. The students are shown the precise areas of the outcrop to look at rather than having to search for field evidence themselves, and broader skills such as map-reading are not developed. Generally, the virtual environment is not as*
*inspiring as being out in the field for real (no matter how hard we try!) and so may have been less enjoyable for students. I know that I was greatly inspired by undergrad field trips and so I think it's a shame that these students couldn't experience that.*". Concerns around inequality, inaccessibility and exclusion decrease from a pre-course percentage of 14% to zero post-course, this is mirrored by a similar decrease in positive comments around accessibility, equality and inclusion 57% to 29%, but
included comments such as: "*Accessibility for all*". Mentions of positive perceptions on the breadth in learning outcomes and experience decrease slightly during the course from 57% to 43%, but included comments such as: "*Adds versatility to standard field skills - i.e. implementation of principles during challenging situations (digital). Also adds a greater focus on digital skills that will become ever-more pertinent as geosciences embraces technological applications.*". The number of comments on the
benefits of virtual field trips remained constant, focusing on aspects of cost, travel and variety of geology: "*independent of weather, distance between outcrops, physical fitness of participants*", "*no travelling costs*", "*opportunity to revisit outcrops when a question comes up at a later point in time.*", "*Access to many different field examples form different field areas showing clear geological features (i.e. can pick and choose and are not restricted to the outcrops within reach of accommodation.*",
"*Large scale perspective that provides context before zooming on details of field data.*". Concerns of staff regarding time management appeared during the course rising from no negative comments' pre-course to 43% post-course, which centered on the length of the course e.g., "*5 weeks is a long time to keep focused and whilst I understand the reasoning for this, I think a shorter time period may have been more beneficial in keeping cohesion*" and "*Perhaps the 5 weeks was a bit long - 4 weeks may have been*
*better*". This is supported by a decrease in positive comments on time management from the pre-course analysis of 43% to 29% post-course.

### 4.2.3 Pre-course perceptions of students and staff

Figure 5c, enables comparison of student and staff perceptions pre-course. The shape of the radar plots for the two cohorts show a similarity in pre-course perceptions between staff and students with negative aspects of the virtual field trip featuring most strongly for both cohorts 88% and 71% respectively. Students and staff appeared to have similar, but relatively low, levels of concern regarding inequality, inaccessibility and exclusion, 19% students and 14% staff. Students were more concerned (63%) than staff (29%) about the negative aspects of online teaching and learning. Staff felt that there would be positive implications for accessibility, equality and inclusion (57%) whereas students barely mentioned this (13%). Positive aspects regarding breadth in learning outcomes and experiences (50% students, 57% staff) and benefits of virtual field trips (56% students, 43% staff) were similar. Concerns regarding time management were zero for staff and 19% for students.

### 4.2.4 Post-course perceptions of students and staff

Figure 5d, enables comparison of student and staff perceptions post-course. The shape of the radar plots shows both similarity and divergence. Negative aspects of the virtual field trip feature strongly for both cohorts 88% students and 86% staff. Students mention inequality, inaccessibility and exclusion (21%) whereas staff do not. Students (50%) and staff (57%) have post-course concerns around negative aspects of online teaching and learning. Perhaps surprisingly students commented on the positive aspects of the online teaching and learning environment more (71%) and were divergent from staff (14%). Staff were more positive (29%) regarding accessibility, equality and inclusion than the students (14%). Breadth in learning outcomes and experience were mentioned by 50% of students and by 43% of staff respondents. Students (57%) and staff (43%) both recognized benefits of the virtual field trip.

### 5 Discussion

The exercise of running the virtual field trip and eliciting perceptions, provided an opportunity to really consider the impact of online course design and delivery choices on student learning and experience. A caveat to our findings is that the global pandemic created a very specific set of circumstances for delivery and engagement of students, and responses will reflect the additional pressures of the time. Irrespective of this, we feel our aim to use the opportunity to reflect on online and virtual field teaching environments and practices to inform future teaching strategies and pedagogy has value. We discuss the findings of our research through the framing of our research questions.

### 5.1 Did the participants and staff perceive that the learning outcomes were achieved?

Before the course staff held several sessions to discuss how best to design and deliver the virtual field trip, these included both consideration of learning outcomes and identification of areas or elements of online learning and virtual field trips that might raise concerns in terms of delivery and learning. Despite not being in the field staff felt in advance of the virtual field trip that all the main learning outcomes could be met, apart from observing rocks in the field. Post-course staff felt that the main learning outcomes were met or exceeded, with medians of 3 or higher for all learning outcomes apart from 'seeing rocks in their context in the field'. There was a recognition that the types of skills and learning outcomes were different and broader in scope than those that might have been learnt on an

actual field trip. This is reflected by positive perceptions around the breadth of learning outcomes and
experience, evidenced by qualitative statements as well as high scores (medians of 4 and 5) in the likert test for data analysis and synthesis. Notably, there was a statistically significant positive shift pre- and post-course for staff for the likeliness of the learning outcome 'interpreting field observations to build a geological history'. So, although skills such as physically taking strike and dip measurements in the field had not been met, manipulating such measurements in an online stereonet package, analysing
larger datasets to build a geological history and digital literacy were, for example, much expanded in comparison to an actual field trip. Other virtual field trips report improved digital literacy as an outcome (e.g., Delacruz, 2019). Students had similar perceptions to staff pre-course about their ability to achieve the learning outcomes in advance of the course, or more positive. Specifically, they were statistically more positive, than staff, in their ability to build and maintain a field notebook and in field sketching
pre-course. The students remained positive post-course, with all medians at 3 or above for the quantitative analysis of learning outcomes apart from for completing sedimentary logs (median of 2 post-course). Students, like staff, also recognised that they had more time to work on data, analysing and synthesising it to expand their understanding and learning, such as thinking about and creating cross-sections that worked with their maps to visualise a fully 3D subsurface space. These types of
skills in critical thinking and analysis are developed through time and are transferable across disciplines and hence in many ways are more desirable than a specific ability to undertake a technique or measurement which can be learnt at any stage.

**5.2 Were effective measures put in place to support the learning outcomes and delivery?**
Staff identified concerns about the students' ability to participate fully in the online field trip in advance of the course. These included internet connectivity, time and space to learn including work and caring responsibilities, as well as issues with engagement and cohort cohesion. The results from online internet tests that the students undertook in advance of the course indicated that student internet speeds were poor relative to data for the whole of the UK and globally at the time of the course. The mitigations put
in place included recording all sessions and providing a printed workbook for students to refer to alongside online course material. The exercises were designed so they could be completed without the need to access 3D virtual outcrop models, with these elements adding value rather than being critical. The pre-course and post-course perceptions of students and staff indicate that these mitigation measures were at least partially successful, although rendering of large 3D virtual outcrop models caused a
problem for some. The workbook compensated, at least to some extent, in terms of exercise completion; although we recognize for student perceptions of inclusion this mitigating strategy could have negative implications. Issues of accessibility and inclusion, most notably around mobility, cost and cultural issues are often thought to be negated by virtual field trips (e.g., Bursztyn et al. 2015) might in fact be replaced by other exclusionary barriers, related to access to high internet speeds, hardware with
powerful processing and the requirement for high-end graphics cards, as recognized by Kelly et al. (2004), Laksana (2021), Pearson and Reddy (2021), Devkota (2021) and Ochieng and Gyasi (2021). For any of the questions posed, there was the greatest difference between the perceptions of staff and students, pre- and post-course, about the impact of caring responsibilities on course engagement. The students felt the impact was low in comparison to staff. Although we do not know why this was, we can

hypothesize that it may reflect staff concerns about both individual student situations as well as their own.

### 5.2.1 Length of Virtual Field Trip

Students recognized the benefits to themselves personally, as well as for the group running the course
over a five-week time period. The length of time was difficult for some as they were concerned with changes to rules and regulations regarding dynamic COVID-19 restrictions, evolving work commitments, and the ability to travel to see relatives amongst other things. We believe that this mainly reflects the uncertainty of the period rather than anything specific to the delivery. Time management was perhaps the most significant issue that resulted from delivery over a five-week period. For students
effectively in lockdown at home there was a tendency to spend a considerable amount of time, beyond that advised on the tasks and exercises. Students felt in some instances that staff had underestimated the amount of time they took for some of the exercises. These negative aspects of time management, alongside the positive aspects of being in control of when they chose to learn are clearly reflected in the questionnaire and statement responses. We believe that the time spent by students, beyond that
expected, resulted in a greater breadth and depth of learning. The extended length of time for delivery also allowed techniques and concepts to embed and skills to develop, particularly critical analysis and synthesis, acknowledged in the post-course scores in learning outcomes. s. But raises conflicting concerns around time management, mental fatigue and student welfare when home working, especially in a lockdown scenario in which the world was all work with little opportunity for 'play'.

### 5.2.2 Recorded sessions

Evidence from the online learning environment and verbal feedback from students indicated that they went back and watched recorded sessions in their own time. This included Q&A sessions as well as formal introductions to the different exercises, to go over material (repeating content) as well as to
access elements that they had missed either through clashes with other commitments or due to internet issues. These positive aspects of the online virtual field trip and learning environment were also reflected in answers to the questions and the free text. The advantages of being able to review recorded material have been evidenced by others across a range of subject areas (Cascaval et al., 2008; Manea et al. 2021).

### 5.2.3 Hardcopy workbook

The workbook was identified by the students as a helpful reference for future learning with a median of 5 pre- and post-course, although staff perceptions of this were more spread, their median score was 4. The workbook was effective for students in terms of allowing them to read and reflect off-screen during
the course mitigating internet connectivity issues, as well as ability to manipulate models etc., whilst reading instructions. This was particularly important for students who were often working on relatively small laptop screens.

**5.3 Did the participants and staff perceive the peer-peer and academic support to be effective?**

Elements of student cohesion and teamwork that result from an actual field trip, alongside peer-peer learning are often considered one of the benefits of face-face learning (Baker and Woods, 2004) and the in-field experience. Staff were concerned that these elements would not be replicated in a virtual field trip. To mitigate, or indeed to try and best replicate these elements students were divided into small groups of five with a PhD student mentor and had drop-in sessions each week to discuss their work and exercises. This helped break down barriers between staff and students and encouraged students to share their work with their peers to discuss issues. Some of these groups worked very well, others were less effective. One group was amalgamated into the other groups part way through the virtual field trip to increase effectiveness this dynamic ability to adapt based on student feedback worked well.

Delivery of the virtual field trip benefitted from a relatively small student cohort that had already worked together in the field and had spent 2.5 years at university ahead of the COVID-19 restrictions. This meant that they were already effectively working well as a cohesive team in advance of the field trip, with their own networks and social media groups which helped with peer support. Students were encouraged to use the chat function in the online learning environment within their small groups and with the whole cohort. However, use of the chat function in the virtual learning environment was limited outside of timetabled live Q&A sessions; although these live sessions were run frequently (minimum daily) throughout the virtual field trip, so extra questions may not have been required.

There was some concern around how effective student support from staff would be in the online environment. Students were positive in terms of many of the aspects of the online teaching and learning environment, the free text responses indicate that students appreciated the large number of staff and student sessions and the ratio of staff: students. As Baker and Woods (2004) describe this level of engagement results in a feeling of immediacy. Actual field trips for a student cohort of 22 may have 2 teaching staff and 1 or 2 PhD student demonstrators whereas there were 3-4 lecturers available at any one time and 5 PhD student demonstrators for the virtual field trip. It was felt, as evidenced by the qualitative responses, by both staff and students that the increased contact time was beneficial to help identify and work through areas of misunderstanding. There was also more opportunity for students who may not normally engage to ask questions of staff and PhD students. This was also seen by a statistically significant shift in the students' responses pre- and post-course that recognized the additional amount of support from academic staff to them as individuals in comparison to an actual field trip.

## 6 Conclusions and Recommendations

The author team, consisting of students and staff involved in the virtual field trip, feel that overall, the virtual fieldtrip was successful in achieving the learning outcomes, based on student and staff perceptions. We acknowledge the small number of student and staff participants and recommend surveys with bigger cohorts to provide results that can be statistically analyzed. Here we have considered broader elements including learning outcomes, peer and academic support and student cohesion. Reflecting on the running of a virtual field trip from student and staff perceptions to inform future online learning and particularly virtual field trips. Based on our findings we recommend consideration of the following elements in virtual field trip delivery:

*Hardcopy Workbook* - the use of a hardcopy workbook delivered to students allowed them to have a tangible overview of the field trip in advance, it provided an easy-to-use set of reference material, enabled students to work off-screen and to evaluate multiple sources of material: online and hardcopy
during a single exercise. It also allowed those with internet connectivity or hardware issues to participate in the virtual field trip and complete exercises solely on paper, although no students were in this position.

*Session Recording* – we recorded all formal sessions, as well as informal Q&As. The students used
these to refresh material as well as to catch up on missed sessions. The only sessions not consistently recorded were small student group sessions with PhD students, as our aim was to make these as informal, relaxed and open as possible. Students were positive about the opportunities available through the recording of online teaching to catch-up on and revise material in their own time.

*Recognition of Challenges* – although we did not use the data on internet connectivity beyond identifying students who might have issues. We believe it was reassuring for students to know that we had considered possible barriers to participation in terms of internet connectivity, space to work effectively, other commitments etc. This helped to build trust and a shared understanding, so that when issues did arise students' felt more able to raise them. The students in comparison to staff did not feel
that caring responsibilities affected their ability to engage in the virtual fieldtrip, but we recognize that this is cohort specific.

*Multiple Interaction Opportunities* – providing multiple interaction opportunities with both academic staff and PhD students throughout the field trip was important. It is also important to recognize that
students will likely also have their own social networks and to build on these.

*Breadth of Learning Outcomes* – our main conclusion is that virtual field trips offer an additional method of training and in many ways complement actual field trips. They can provide opportunities for greater and different interactions with staff that are not possible when those same staff are also dealing
with the logistics of an actual field trip. They provide opportunities for greater synthesis of data and development of critical analytical skills over multiple geological field areas that is hard to replicate on an actual field trip.

The radar plots reflect the nuances of positive and negative aspects of virtual field trips. We note that
both pre- and post-course, for students and staff, negative aspects of the virtual field trip dominate comments. But that these are countered by positive comments on aspects of virtual field trips and online teaching and learning and perhaps most notably the breadth in learning outcomes and experience. We can learn from this drawing-on the findings to inform future design and delivery of virtual field trips in a blended-learning environment to expand and develop their positive aspects.

**Author Contributions**
CEB lead the virtual field trip, conceived the research, led on staff discussions around learning outcomes, barriers and mitigation measures, deigned and deployed the questionnaires, designed the

initial data analysis protocols and completed an initial data analysis, lead on writing the manuscript.

JHP acted as PhD student demonstrator on the virtual field trip and input into initial discussions on course design and delivery, coding discussions of the qualitative text, redrafted all figures for final publication, collected the data for Figure 1 and commented on the final text draft. LK acted as PhD student demonstrator on the virtual field trip and input into initial discussions on course design and delivery, analyzed and created initial box and whisker plots for the quantitative data and inputted into

coding discussions of the qualitative text, co-authored the initial draft text for qualitative statement analysis. SRL was a student on the virtual field trip, they input into data compilation, coding of the text-response statements, co-authored the initial draft text for qualitative statement analysis. MZS and TKG were students on the virtual field trip they conducted the initial coding of the qualitative text and input, and drafted some of the methods text, and constructed initial radar plots. MB was a student on the

virtual field trip they input into initial discussion and helped with qualitative statement coding.

## Data Availability

The underpinning data for the likert survey and the Mann-Whitney U test results are available https://doi.org/10.20392/89c5596a-7d0b-44fa-bd1a-f40b18f45b94


## Conflicts of Interest

There are no conflicts of interest identified.

## Acknowledgements

We acknowledge the input of all students and staff that took part in the virtual field trip and contributed to the research by completing the surveys. The virtual field trip academic staff were Clare Bond, Rob Butler, Malcolm Hole, Colin North, Adrian Hartley, and Ian Alsop who were involved in discussions on learning outcomes course design and delivery modes. The non-author PhD student demonstrators were Tom Theurer, Bartosz Kurjanski and Sophie Berhendsen who provided feedback on small group

sessions.

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

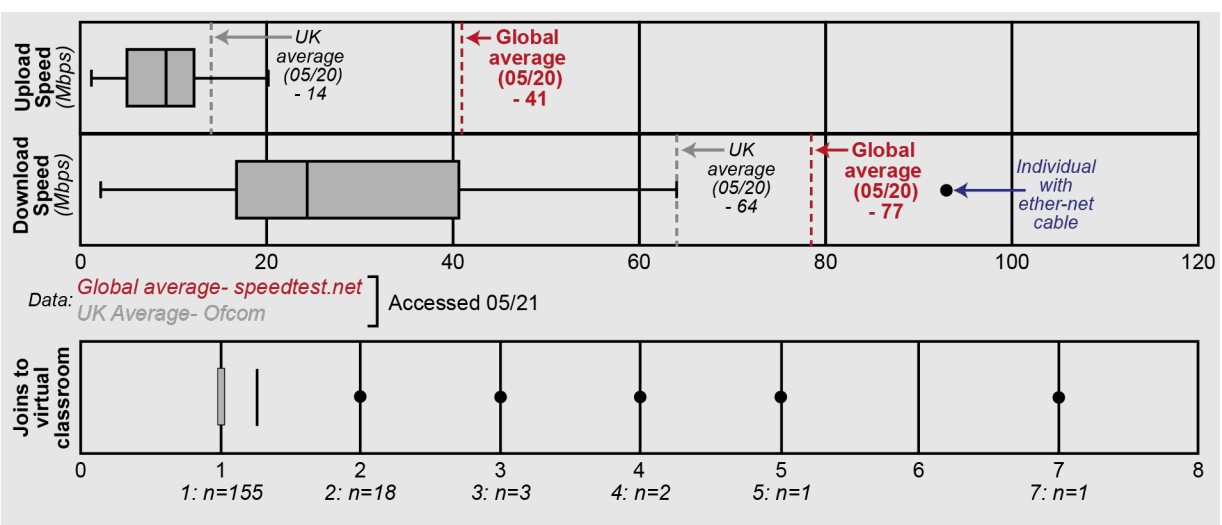

**Figure 1: Internet speed and connectivity data. a) Student internet upload and download speeds presented in Mbps, and plotted as full and interquartile ranges, with averages marked. The student cohort data are compared to UK and global averages for the same time period as the virtual field trip (May 2020). Note that the one student with an ether-net cable plots as an outlier for download speeds. b) Data from the online student classroom that summaries the number of joins per session to the virtual classroom to 22 live online sessions over the five-week virtual field trip.**

| Positive or Negative | Code Name | Code Descriptor |
|---|---|---|
| + | Accessibility, equality, inclusion | Students gain the same experience regardless of their life situation, level of fitness, and other capabilities |
| + | Aspects of online teaching and learning | Set sessions for questions, greater access to staff, group working, demonstrator interactions. Available reports of student performance, staff and demonstrator input. |
| + | Aspects of the virtual field trip | Ability to manipulate between micro- and macro-scales, different outcrop perspectives, visit multiple outcrop locations irrespective of distance between them, in a single trip. |
| + | Time management | Ability to work at one's own pace and to plan the spread of workload over preferred timespan. |
| + | Breadth in learning outcomes and experience | Students gain and practise a wide array of skills (both filed, digital and other transferable) e.g., Learning new software (Lime, Virtual Outcrop, Google Earth Pro, Stereonet) which help to visualise geological concepts. Experience relevant for future career paths and learning to work with other peoples' data. |
| - | Inequality, inaccessibility, exclusion | Possible distractions, implications of students' workplaces on their performance. |
| - | Aspects of online teaching and learning | Lack of easy social interaction to build cohort and team ambiance. Limited 'direct' peer learning experience, nature of interactions not continuous, working from home feels overwhelming and isolating. Hard for staff to get a feel for student learning and understanding. Issues of motivation and wellbeing. |
| - | Aspects of the virtual field trip | In field observation, measurements and map reading skills not developed. |
| - | Time management | Duration of exercise completion much longer than suggested. Five weeks too long, took over our time. |
| - | Software, hardware and WiFi problems | Issues of working online (internet quality, performance of hardware and software). |

**Table 1. Summary of codes and descriptors for the qualitative text analysis. Colored for positive (green) and negative (red) aspects.**

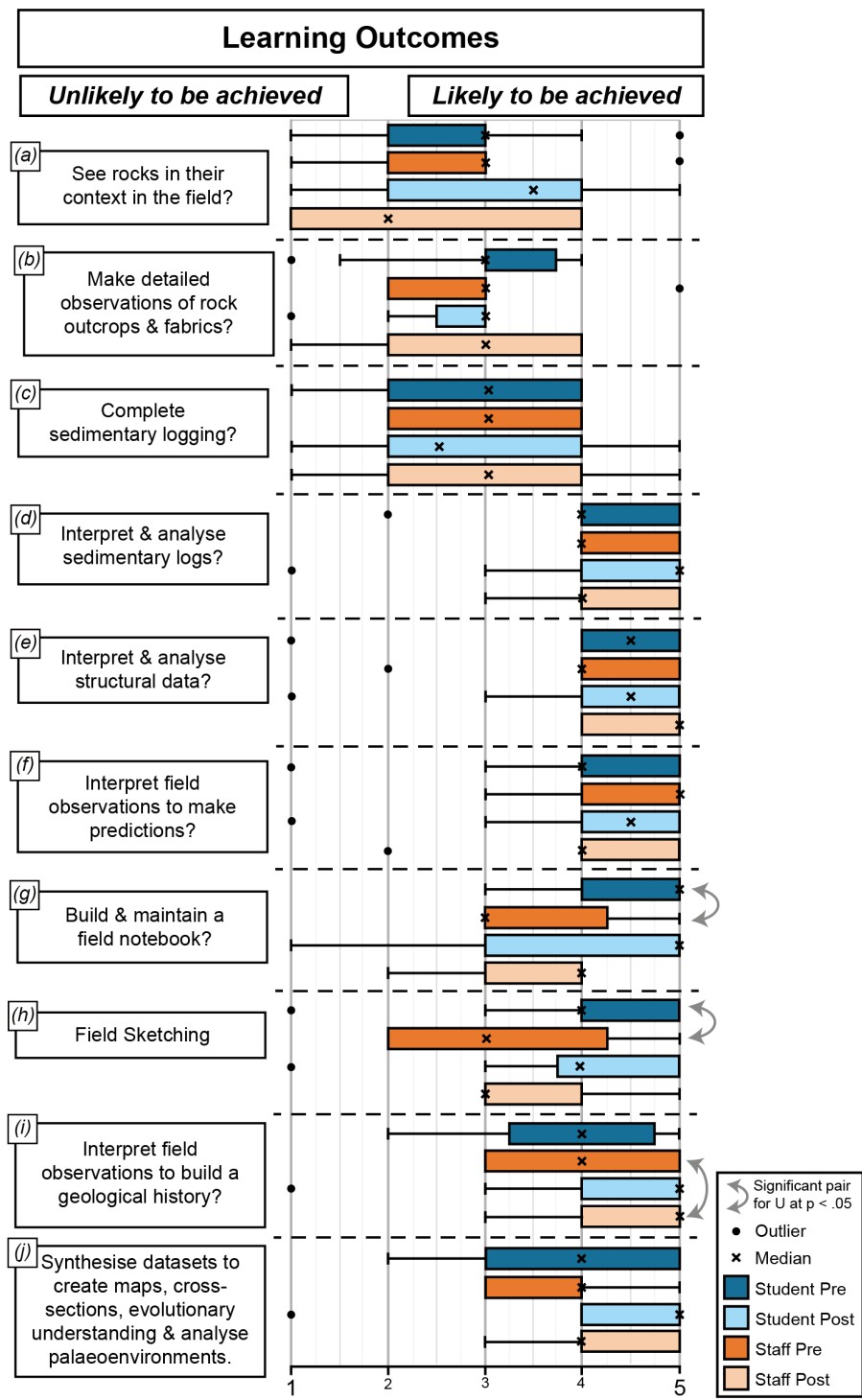

Figure 2. Quantitative responses to learning outcome statements. a)-i) Summarise staff and student responses on a scale of 1-5, where 1 is unlikely to be achieved and 5 is likely to achieved

**for a series of statements on learning outcomes. Responses are grouped by statement and presented as pre-course and post-course for the student and staff cohorts.**

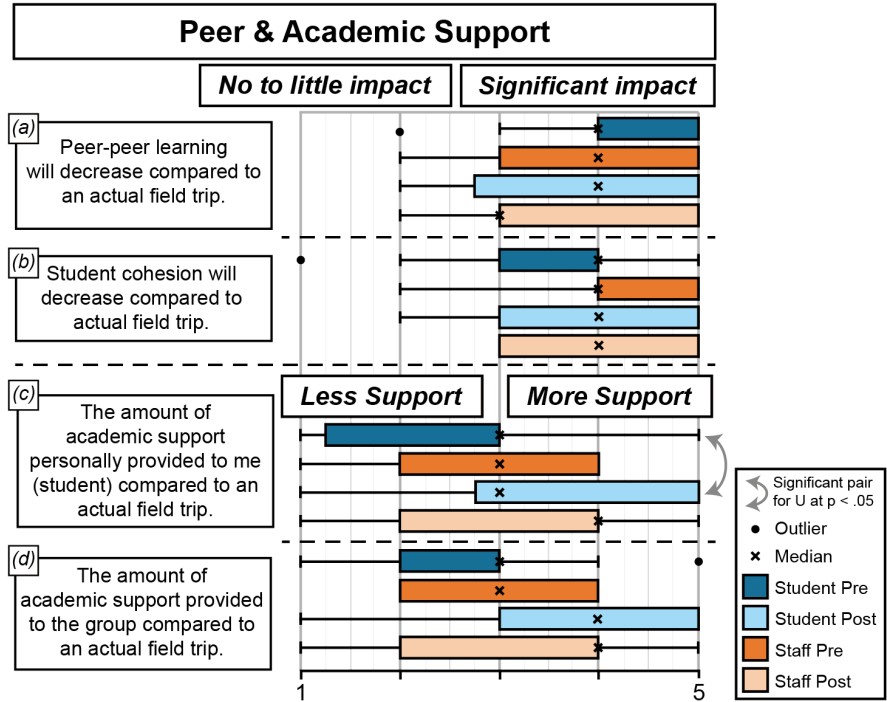


**Figure 3. Quantitative responses to peer and academic support statements. a) & b) Summarise staff and student responses on a scale of 1-5, where 1 is no to little impact and 5 is significant impact on statements related to peer-peer learning and student cohesion. c) & d) Staff and Student responses to statements on support level where 1 in less support and 5 is more support.**

**Responses are grouped by statement and presented as pre-course and post-course for the student and staff cohorts.**

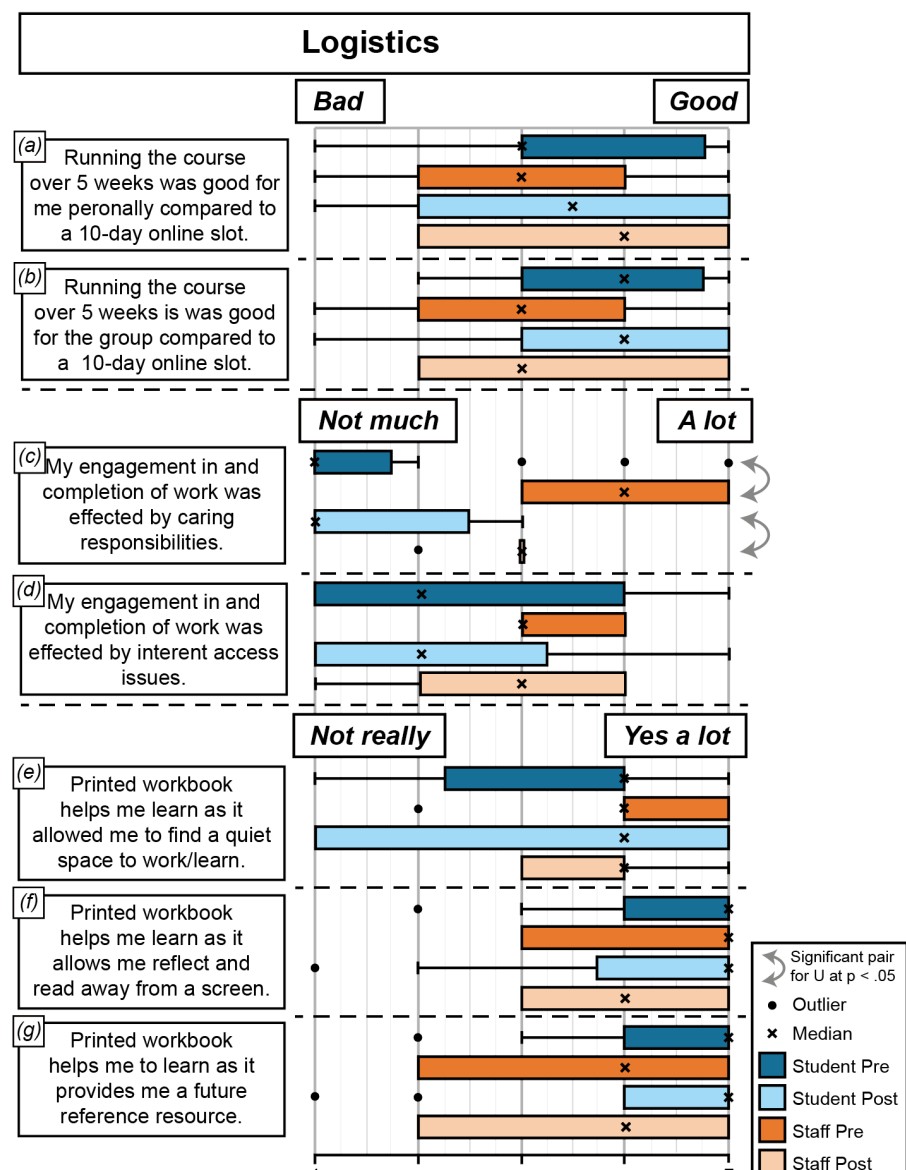

**Figure 4. Quantitative responses to logistics statements. a) & b)** Summarise staff and student responses on a scale of 1-5, where 1 is bad and 5 is good for statements related to the length of the virtual field trip. **c)** Staff and student responses to statements on the impact of internet access issues on completion of the course, where 1 is not much and 5 is a lot. **d)-f)** Staff and student responses regarding the usefulness of printed workbooks, where 1 is not really and 5 yes a lot. Responses are grouped by statement and presented as pre-course and post-course for the student and staff cohorts.

| Code Title | | Students | | | | Staff | | | |
|---|---|---|---|---|---|---|---|---|---|
| | | Pre | | Post | | Pre | | Post | |
| | | (%) | Frequency | (%) | Frequency | (%) | Frequency | (%) | Frequency |
| Accessibility, equality, inclusion | +ve | 13 | 2 | 14 | 2 | 57 | 4 | 29 | 2 |
| Inequality, inaccessibility, exclusion | -ve | 19 | 3 | 21 | 3 | 14 | 1 | 0 | 0 |
| Aspects of the online teaching & learning | +ve | 19 | 3 | 71 | 10 | 14 | 1 | 14 | 1 |
| | -ve | 63 | 10 | 50 | 7 | 29 | 2 | 57 | 4 |
| Aspects of the virtual fieldtrip | +ve | 56 | 9 | 57 | 8 | 43 | 3 | 43 | 3 |
| | -ve | 88 | 14 | 86 | 12 | 71 | 5 | 86 | 6 |
| Time Management Positive | +ve | 25 | 4 | 36 | 5 | 43 | 3 | 29 | 2 |
| | -ve | 19 | 3 | 36 | 5 | 0 | 0 | 43 | 3 |
| Breadth in learning outcomes & experience | +ve | 50 | 8 | 50 | 7 | 57 | 4 | 43 | 3 |
| Software & hardware/WiFi problems | -ve | 25 | 4 | 36 | 5 | 14 | 1 | 0 | 0 |
| | | | n = 16 | | n = 14 | | n = 7 | | n = 7 |

**Table 2. Summary of qualitative text coding responses. The table figure shows the ten codes and frequency and percentage of coded occurrences, this data is plotted in figure 5.**


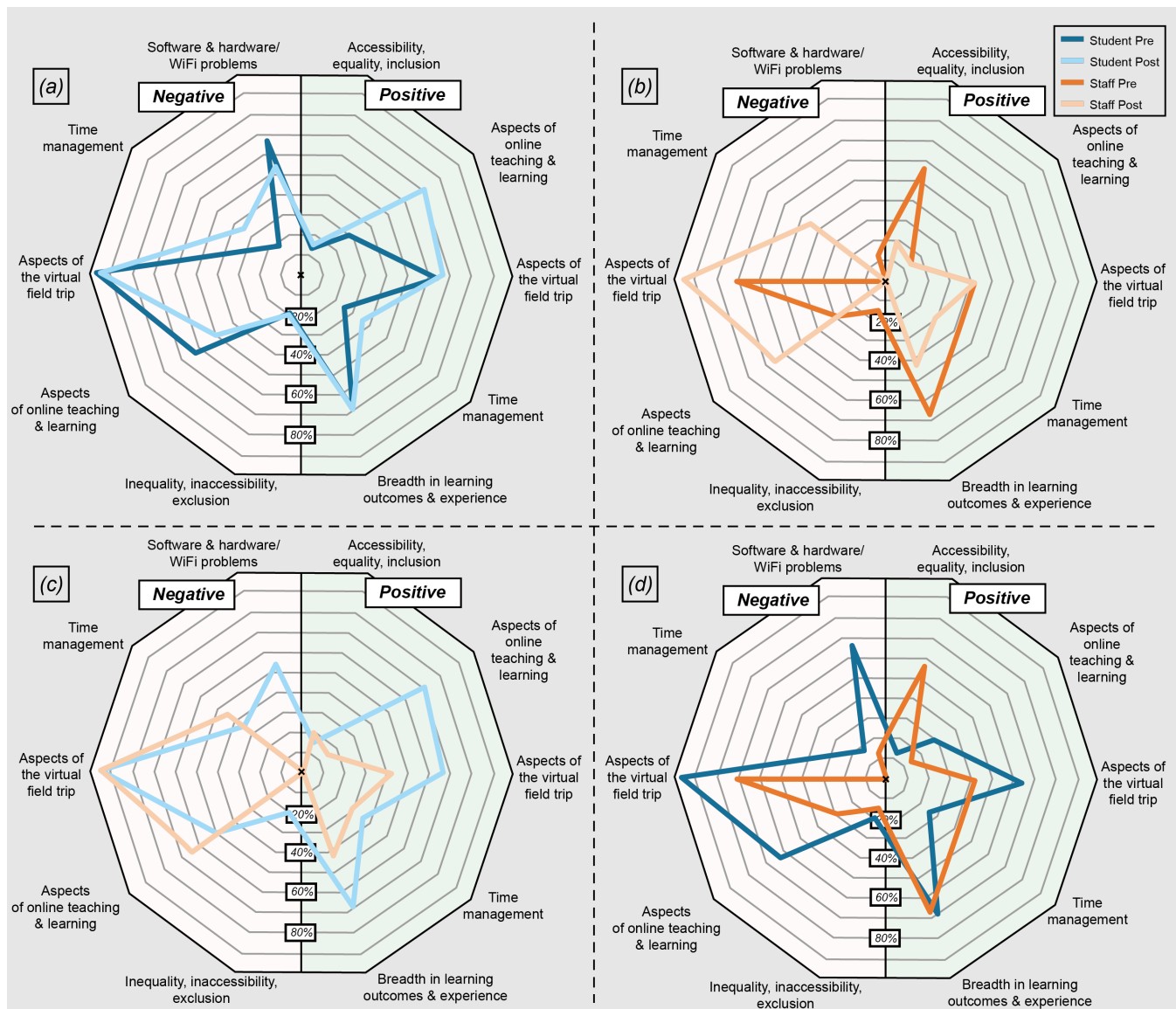

**Figure 5. Radar plots of student and staff perceptions derived from free text responses to questions. a) Student perceptions pre-course (n=20) and post-course (n=14), b) Staff perceptions pre- and post-course (n=7), c) Comparison of student and staff perceptions pre-course, d) Comparison of student and staff perceptions post-course. The plots are split vertically with negative elements on the left and positive on the right, positive and negative equivalents are plotted opposite each other, except for two independent codes.**