# Peer review of "Learning outcomes, learning support and cohort cohesion on a virtual field trip: an analysis of student and staff perceptions"

_Geoscience Communication, 2021_

## Author Response (AR1)

**We have added commentary on changes and responses in red.**

RC1 Comments

General comments

This manuscript describes a considered approach to the delivery of a virtual field trip during the COVID19 pandemic. The authors collected questionnaire data to evaluate how the students and staff perceived the strengths and weaknesses of the approach. I strongly recommend major revisions that would allow the team to reorganise key information, address questions raised by the reviewers, consider different approaches to the data descriptions appropriate to the datasets, and rewrite the discussion section to be evidence-supported and literature-supported. The main issue is that the reader cannot distinguish the difference between author (instructor?) reflections and claims supported by the data. These need to be clearly distinguished for the reader.

RC1. Does the paper address relevant scientific questions within the scope of GC?

Yes

RC2. Does the paper present novel concepts, ideas, tools, or data?

The virtual field trip design and delivery considerations is potentially novel, but without presentation of the literature it is hard to judge this aspect.

New literature added.

RC3. Are the scientific methods and assumptions valid and clearly outlined?

No. The methods section needs some additional information to be improved. There are few (no?) assumptions or limitations described by the authors.

Methods have been revised (see details below).

RC4. Are the results sufficient to support the interpretations and conclusions?

No. The data presented needs to be amended to accurately represent the data collected. Additionally, the discussion section needs to be rewritten to clearly reflect assertions that are evidence-based, versus opinions of the instructors.

Completed.

The quantitative results need to be amended to valid statistically descriptions of ordinal data. The qualitative results are quantified, rather than including rich text to show the lived

experiences of respondents. The qual section would benefit from direct quotations to help qualify the data.

Quotations added.

RC5. Do the authors give proper credit to related work and clearly indicate their own new/original contribution?

RC6. Does the title clearly reflect the contents of the paper?

Yes, but it is overly long and complex. I think the title needs to be edited to reflect the major findings of the research.

We think the title reflects the paper.

RC7. Does the abstract provide a concise and complete summary?

Somewhat. The abstract needs to have a clear distinction of the main design features of virtual field trip (what makes this VFT unique to other VFTs?) and clearly outlined results of the study was (evidence supported)

Data has been added to the abstract.

RC8. Is the overall presentation well-structured and clear?

The structure is OK, but course design information is peppered throughout. The content related to course design needs to be moved together. There needs to be only results in the results sections, and the discussion needs to explicitly link to the data in the results.

Course Design section has been restructured along with the results and discussion.

RC9. Is the language fluent and precise?

Yes, but the manuscript requires a copyedit and a check for grammar throughout.

Checked.

RC10. Are the number and quality of references appropriate?

No, there is a deficit of appropriate literature in this manuscript

Further literature has been added.

Specific comments

Abstract

All data and results need to be clearly written and evidence-based (See later comments in review). Readers should understand the difference between an instructor(author) reflection and a claim supported by data.

We feel this is now clear.

1 Introduction

Defining the common acronyms for VFE's, VFT's, or VLE's might be helpful here for the reader (if you plan to use one throughout your piece).

We just use Virtual Fieldtrips

There are published materials comparing traditional and virtual field experiences in the geosciences prior to COVID. I would suggest looking again. Namely, take a look GSA abstracts from 2009 onwards for authors reporting on comparisons (those authors will have peer reviewed materials that can support your work). One such work looks at student field experiences in a videogame vs. real life (Dohaney et al 2012).

We have added further literature.

I would recommend clearly writing out the research questions for the reader rather than alluding to "your focus" or weaving together your "we are interested in" statements. Having clearly stated RQs allows the data and results to be judged against those RQs, and the strength of your evidence more accurately weighed.

We have added in clear research questions.

Line 66 What do you mean by learning ability? This is a vague term, and would benefit from specific language like "learning gains", retention of information, or other more specific language that is useful in education research

We have supported this with references.

Line 67/68/69/70 Reference Novelty Space research by Orion

Is going to a new physical location, and encountering new people and places of a similar impact to students' novelty space as trying new online software?. I would argue otherwise, but happy to be incorrect if there is research that backs this claim. Look for research on novelty space and educational technology to support your ideas here.

We reference Orion and Hofstein, 1994 and explain what we mean in this context.

Line 74 Is learning to be measured in this article? If so, foundational literature will be needed to establish your measure for learning. If it isn't measured, then you need to be clear that it is student perceptions of learning (not actual measures of learning) being captured.

Rephrased.

Line 79 online questionnaires? A survey is different from a questionnaire

Amended.

2 Course Design

In this section you tell us about your considerations when designing the curriculum, but you don't actually tell us what the final design was (in a clear way – it's woven together). A course design section should tell us what the design was/is.

We cover the key elements of the list below that are relevant to the study, and have reframed the course design section.

To support replication or comparisons of the curriculum, it is recommended to include responses to the following checklist of questions:

Did the learning take place in the classroom (lecture), laboratory, field or other type of learning environment?
Was the mode of learning and interaction with students predominantly in person, online or in blended mode?
Who is the curriculum appropriate to? Describe the level and demographic information (if appropriate, and if ethical approval was granted)
Is this learning appropriate to other groups of students not described in this research? (e.g., appropriate to all levels of education, etc.)
How many learning activities are you describing?
What is the duration of the learning activities?
What types of learning activities are you describing?
What academic disciplines would this curricula be appropriate for?
Are the students assessed? Describe the assessment.
What type of student interactions are there? Do they work independently, paired, or in groups?
Where can the wider teaching community access the full curriculum detail? Are there additional curricular materials that could be provided to the reader to support the learning and teaching process? (Provide a permalink/doi to a source of information, or within an appendix)
In what country(ies) did the learning and research take place?
What institution(s) participated in the learning and research? How many were there?
What kind of institution(s)? (Imagine that someone reading doesn't know the difference between types of institutions in your home country)

What course(s) or class(es) did the learning take place? What topics are being taught in this course?

Check your writing against the above to make sure that this information is covered in your description.

See comment above.

Line 96 What are the learning outcomes of this virtual curriculum? They should be listed for the reader. (Are these the questionnaire questions? If so, be explicit and include them in-text)

Added these.

Line 100 what do you consider cohort cohesion? Connect this to educational literature. Do you mean student relating to one another, forming social bonds? Do you mean students working together as a team? Whatever best fits your ideas here, please present literature to support it and define it.

Added references to support this.

Line 104 Is data from the initial questionnaire included in this article? If so, this is considered results. Those results should be included in the results section, rather than a course design section. Revisit the prompts above for what specific information should be included in a course design section.

Included in the results.

Line 110 Any dataset must be described in terms of how many people were invited to participate and how many responded to the questionnaire (in this case). All of this information is included in the methods section (Section 3?)

This information had now been added.

3 Eliciting Perceptions

There is no description of the research design approach (i.e., your philosophical perspectives, paradigm, and the specific methodology applied). Is this a case study? Is this a mixed methods study? Describe specifically the research design used in this research.

Here are some specific questions that should help provide useful information:

What are your research questions in this study?
What is your approach to the research?
What research methodology and paradigm are you using?
What research method(s) are you using in this study? Describe in detail.

What empirical or anecdotal evidence have you gathered to support your research questions? What kind of evidence and data was gathered? How much evidence/data was collected?

What kind of phenomena are you attempting to measure/characterise? (student learning, perceptions, attitudes, performance, behaviours…etc?)

Describe the recruitment, sampling, and approved ethical human research process of gaining consent for the research

We have considered these points and added more information as appropriate.

Who are the students? Here are some questions that should provide useful information:

How many students/participants are there? (Important note: please indicate the difference between how many students participated in the curriculum, vs. how many consented to participating in the research)

What level of education are they currently undertaking?

What year of study are they current undertaking?

What degree programme or majors are they undertaking?

- What key demographic information can you include that helps us understand who they are and how that relates to your research questions/curriculum outcomes? (Gender, age, race, ethnicity, languages, disability, prior academic experience, nationality, immigration status, and/or social class, amongst other factors)

We have ensured the relevant information is presented. Our study did not attempt to consider demographic information, and this was not included in our ethics.

3.1 Questionnaire Design

You should incorporate the literature on why the use of those specific question formats are needed here. Why Likert scales? Why open questions?

We have added to this section.

Line 114. Very important: Was ethical approval to conduct this research approved by your institution? Any and all data collected from human beings needs to be approved by a higher education institutional body. Your university will have specific guidelines that need to be followed on how data is collected, stored, and the nature of inquiry (what kind of questions did you ask, and what risk and benefits are there from people participating in your study)

Yes

Line 115 Who was invited to participate? Were there any exclusion criteria? Specifically how were the participants invited to participate (email, etc?).

We have included this information.

Line 115 Describe the response rate to all of the questionnaires.

Included.

Line 115 How was informed consent to participant in the study gained? In writing?

Yes, you can see this in the supplementary material.

Line 116/126 Specifically when were the questionnaires administered (timing of the course)?

We have added in information.

Line 121 Were the questions (statements) your own design or did you gather them from existing validated instruments?

Yes, see section on survey design and link to learning outcomes.

Line 121 Were the questionnaires validated for content validity? Did you test the questionnaire prior to use?

Questions were developed by the lead author.

Line 135/6 Given the content of the questions, did you research other researcher's questionnaires to establish valid statements in these themes (LOs, support and logistics)?

Not sure what the reviewer means?

Supplementary Material

Are Qs 1-3 open questions? Provide instructions on the supp material sheet

All included.

This is the student version of the questionnaire. Are there any additional questions that were included for staff?? If so, include at the bottom with something like: "Additional questions were asked such as xxx, yyy"

All are now included, for clarity and so they can be reproduced exactly.

You use a variety of 5-point scales, that may/may not elicit changing beliefs. For example, a scale of 1 (bad) and 5 (good) – what does 3 look like? Sort of bad and sort of good?

The pre questionnaire (about access) should also be included here, if any of the data is discussed in the article.

This is now included.

3.2 Questionnaire Analysis

Line 141. What kind of coding were you doing? Please describe and reference the coding method used. Thematic analysis? Constant comparative analysis? Content analysis? Etc.

We explain the coding analysis and have included code descriptors.

Line 147. The key themes are results – suggest moving those to the results section.

Moved.

4 Internet access, student availability and other issues

What is the overall aim and purpose of this section? Is this a course design section or a results section?

Moved as suggested above.

Overall, there is a blend of information presented in this section. I interpreted section 4 to be a results section. A results section should describe and report on the data presented. There is a combination of course design materials, author (teacher?) reflections, and data combined into one. I think it should be clear to the reader which is which. In my experience, all course design information is presented in a specific section, and then the results of survey information is presented showing support for/reflection on/in critique of the course design. This way the information is clear to the reader, allowing them to judge for themselves. I would strongly suggest a reorganisation of the information in this way.

4.1 Internet access

Line 158-159 overly complex sentence, please amend

Amended

Did you have permission from your institution to use the learning analytics described in this section? It wasn't mentioned in the methods section about the tracking of student sign-ins, and other data described here.

Yes, now described in more detail.

Figure 1 is really clear and effective.

Line 170+ . I think it's important that the content of these sections is results from the surveys and the internet data. Any course design information should be presented in the course design section prior. Try and separate out the types of data: course design elements, survey data, and internet data, etc. so it's clear to the reader.

Moved as suggested earlier.

4.2 Student availability and other issues

Many claims here not supported by data (or is it?).

Line 183/184. What is this statement based on? How do you know the student availability? What data supports this? Are these anecdotal events from students? Try and organise information supported by the data so that it's clear to the reader what the data says, versus what you say (the authors observations/interp of the data).

We asked about student availability in the pre-course questionnaire (see supplementary material).

5 Perceptions

5.1 Quantitative statements

Line 200. How many staff and how many students responded to the questions? (put in text here)

Now included.

You use a 5-pt scale for most of your statements. Revisit how these data are conventionally analysed. Likert scale data (ordinal data) is most commonly approached differently than interval data.

We have referenced others who plot data similarly and have calculated medians as suggested.

Why did the authors use averages to describe the central tendency in your quantitative responses? Explain your thinking and choices when using central tendencies with regard to ordinal data. Medians are more appropriate here – what does the average of Strongly agree and Likely to be achieved and less likely to be achieved?

We have calculated medians.

What statistical tests did you apply to the two groups of data to ascertain differences? Are your cohorts big enough to determine differences? A difference of averages is just that – it's a descriptor but not a conclusive results. You need to do paired comparisons of the groups (staff and students). It is not appropriate to apply t-tests to ordinal data, and so is more appropriate to use Mann-whitney and Kruskal Wallis tests to these data. Take a look at MW and KW tests.

We have now completed a Mann-Whitney U test. The results are now included in the paper and the U test is available in the data file.

Figures 2-4

Are box and whisker plots the best way to represent ordinal data? It is not the most common approach to these types of data. I worry it may be misrepresenting it. Have a think about how else this data might be more accurately represented.

We have referenced others who have plotted similar data in this way and have calculated medians as suggested and completed a statistical test.

There are copy editing errors (spelling) within, please amend

Checked through.

5.1.1 LOs

Line 210. A reminder that the reader doesn't know the learning outcomes, so we can't really situate ourselves for what these statements mean with regard to how effective the virtual field trip was.

We have added Learning outcomes.

"Post-course, the IQR increased for both cohorts, 2-4 for students and 1-4 for staff. The average student response increased to 3.2 and remained fairly constant for staff at 2.6." – Is this the best way to describe changes to the data?

Revisit how pre and post-changes to survey data (with ordinal data) are normally characterised. If we compare groups (pre- vs. post; or staff vs student) we need to conduct statistical tests to do so. See my comment from above. Movements of the range does tell us something, but it needs to be communicated accurately (minium, maximum) rather than using IQR.

See comments above.

Broadly, the descriptive text here is difficult to read. Using a table to show the pre, post, median, and groups (students, staff) would create a shortcut for the reader. Then the writers can highlight the key CHANGES and the key DIFFERENCES between the groups – rather than spending time describing the numbers.

The box and whisker plots show this and we have highlighted the statistical significant elements in the figure.

What is positive shift and what is a negative shift. Take a look at other research on how that is described and quantified in text and in plots.

5.1.2 Support + 5.1.3

All the same comments as above. All quantitative data needs to be revisited for how it is calculated, described and written about in-text and in the figures.

See comments above.

5.2 Qualitative statements

Line 269. Long text answers – open response questions?

yes

Line 269 – if the questions are to be described in detail, I suggest providing the full questions in text for the reader

We do in many places and all the surveys are now available in the supplementary material.

Line 275 – if open response information alludes to the previous quant questions, it is fair game for the researchers to incorporate that text into the sections about those statements. Alternatively you could reorganise your Results section 5 around Themes (recommended) rather than on the "type of response" (qual vers quant).

We decided to keep the different types of responses separate.

Sections 5.2.1., 5.2.2, 5.2.3, and 5.2.4.

Instead of treating the qualitative data as rich text, you've opted to apply quantitive methods to the qual data. I would include a reasoning of why you did this in the methods section.

To identify common themes, and evidence these.

Why not provide the codes and themes and include specific quotes that exemplify those themes?

We have included this now.

I find the text in these sections to be summaries of what is provided in Figure 5, instead of providing rich descriptions (Geertz) of what the participants felt, thought, and reflected upon the course and it's design.

We have now included rich text examples.

What do we gain (as a reader and community) from the quantaitive results here? If the authors decide to include quantitative assessment of these data, a bulleted summary might be more effective.

Figure 5

I do like adding the positive or negative framing of the results here. The % needs to be described in the caption for the reader – what does 20% vs 80% mean?

The positive and negative framing fell-out of the analysis.  We have amended the caption to include information on no. of respondents.

6 Discussion

There are two major flaws with the discussion section, as it is currently. First, the claims made about "what was found in this study" are not explicitly linked to evidence. And second, there is no linkages to the literature. In a discussion section, the authors need to connect their research to the existing literature. There is significant work done on educational technology and student and staff perceptions of virtual field experiences. You currently do not include any meaningful literature.

All claims must link to the evidence. It's not enough to paraphrase the results, the reader needs to see the connection. There are many claims made that don't explicitly link to the evidence provided. You cannot make claims that you don't have evidence for (e.g., students learned more from the virtual field trip – you didn't measure learning, etc.). Try and create some transparency to the reader about what are reflections from the authors (instructors) and what are discussions of the evidence (supported by data).

We have added to the discussion.

6.1 Were learning outcomes met?

Think about use of academic paragraphs in this section. When to start, when to stop, etc.

Line 328. Is this true? How do we know the LOs were met? Back this statement up (link directly to) the evidence.

We have rephrased this.

Line 328 recognition by whom? Link to evidence

Line 330 Sign-posting of evidence is needed throughout. E.g., reduced negative perceptions of online teaching (Statement 5, Fig 3), etc.

We have linked evidence now.

"Students also recognised that they had more time to work on data, analysing and synthesising it to expand their understanding and learning" – where do we learn this? Is this referring changes in perceptions of time management?

We have now added in rich- text to support.

Line 338. If you include anecdotal information, be sure to be clear that the reader is aware of this

6.2 Were pre-identified concerns lived out, and were mitigations measures effective?

This section has a good balance of including links to evidence and discussion of the results.

How do you know which mitigations were more or less effective? (hint, unless the question was specifically asked, you don't). You can assume, but you don't have data to support it.

Ok

6.2.1

Line 359. Do you have evidence that students spent more time on tasks than normal?

See rich text

Line 363. Do you have evidence that students had a greater breadth and depth of learning? Did you measure learning?

This is supported by perceptions of the staff and students.

6.2.4

Line 378. Did you get permission to use the students grades in this study? If you didn't get informed consent to use their scores/grades, you can't allude to it in the research.

Have removed.

6.3 Did the virtual field trip develop student cohesion and peer-peer learning?

Line 387 – benefits of in-person field experience (reference needed)

Supported by rich text.

Line 388 – Mentioning course design aspect here (whereas that should be mentioned in the course design section).

Amended.

Line 390. Do you have evidence that groups helped students break down barriers? Do you have data that describes how the student grouping went? Any qualitative data describing student experiences?

Rich text

Line 395 – Instructors' reflection that students already worked well as a team? (any evidence to support this?)

We have made it clear that this is a reflection.

6.4 Was academic-student support effective?

Line 405. "It was felt" – how could you rephrase to accurately represent the data?

We feel like this covers the data, we make it clear that this paper is about perceptions.

Line 407. Any evidence that students who normally don't' engage much were engaged?

Added in reasoning for statement.

7 Conclusions

What are the major findings of this research study? You don't summary the key findings here, and I think the manuscript would be much improved with this important information.

Line 409. You didn't present student grades (and I suspect didn't get informed consent to use those grades?). What are some other indicators of success? (why not lean back onto the data and results that you do have to make a claim about what success looks like here?

Reference to grades removed.

Starting Line 413 – recommendations could be in a bulleted list

Ok

Do you have data to support your recommendations?

If not, say explicitly. "These are instructor recommendations for how to support students in virtual field trip success:" and then add in (explicitly) where the data you did collect supports your assertions.

We have done this.

New course design elements are included in the conclusions section. My recommendation is that no course design aspects are included in the results, discussion and conclusion sections. These elements should be well described in a course design section early in the piece and then alluded to later in these following sections.

We are not sure what new course elements the reviewer is referring to?

Line 436. Do you feel that the current manuscript communicates students negative comments accurately? I sense that you've glossed over the critiques. How could you better incorporate their views (even the negative ones)?

We have included positive and negative rich text statements.

Technical corrections

Check for comma punctuation throughout your piece. They are often missing, or overused in some sections. Check for use of oxford comma (or not) throughout, consistency

Line 47 Use of comma with e.g.,

Line 54 Do you mean equity (not equality?)

Line 323 … Were the learning outcomes met?

Line 342 … were our mitigation measures effective?

Checked.

CC1 Comments
This is a great reflection on online and virtual field teaching. The manuscript is clearly written and well-structured. I applaud the authors' effort to evaluate key issues related to carrying out a geoscience virtual field trip including internet connectivity, student and staff perceptions on learning outcomes and cohort cohesion. Lessons learnt here will certainly inform future virtual field teaching. A couple of questions and comments:

1. Key demographic data on students and staff are missing including the number of participants completing the questionnaire. If these data were collected, please consider including them.

We have added in the numbers -this was an omission.

2. Ethics and consent - Please include relevant information (in accordance with GDPR) on the ethics assessments and the consent forms used for this study.

Added in.

3. There are 11 learning outcomes (bullet points under question 3) in the questionnaire. Some of these outcomes are missing from Figure 2 and the discussion section. Why is that? Could you provide an explanation?

We have included everything in the revision.

4. Statistical information - It's not clear to me if your results are statistically significant, especially when the number of participants is not reported (and the number of staff is low). Please explain what statistical tests you 've

performed including the corresponding values (in the text and in your figures, if possible).

We have completed a Mann-Whitney U test – details of this are now included in the paper and the test results are available in the data file.

5. Can you include the questionnaire used for testing students' internet connection speeds in the supplemental information?

Yes now included.

6. It is stated that "the answers of some participants might have been influenced by attendance in the initial week's session prior to questionnaire completion". Can you provide more information on this session and how it might have influenced participants' answers?

We have now clarified this section.

7. Was there a reason for the questionnaire being anonymized? This prevented the authors from matching pre- and post-assessment results for individual participants. One way to match an individual's pre- and post-assessment answers (while remaining anonymous) is to ask participants to use a code name (known only to them) on both surveys (e.g., see: https://gc.copernicus.org/articles/4/281/2021/gc-4-281-2021.html).

We did not have a long time to set-up. This would be a good recommendation for future work.

8. "Students were unsure as to the usefulness of a workbook in terms of "finding a quiet space to work" in advance of the course". This statement, as phrased here, was also unclear to me. Could it be that students didn't understand what they were being asked to rate? Perhaps this statement can be phrased differently?

We have no data on understanding, but it is a useful comment if the same survey was to be run again.

9. Great figures, clear and effective!

[Figure]
 ☺

---

## Author Response (AR3)

I have been through the comments in the PDF and made the changes highlighted.
I have also updated the doi link to the data.
Best wishes
Clare